# Personalized Nasal Protective Devices: Importance and Perspectives

**DOI:** 10.3390/life13112116

**Published:** 2023-10-25

**Authors:** Thinh To Quoc, Ildikó Bácskay, Pálma Fehér, Ádám Pallér, Boglárka Papp, Krisztina Bíró, Zoltán Ujhelyi

**Affiliations:** 1Department of Pharmaceutical Technology, Faculty of Pharmacy, University of Debrecen, Nagyerdei Sqr 98, 4032 Debrecen, Hungary; toquocthinh_93@yahoo.com.au (T.T.Q.); bacskay.ildiko@pharm.unideb.hu (I.B.); feher.palma@pharm.unideb.hu (P.F.); adampaller@gmail.com (Á.P.); papp.boglarka@pharm.unideb.hu (B.P.); 2Doctoral School of Pharmaceutical Sciences, University of Debrecen, Nagyerdei Sqr 98, 4032 Debrecen, Hungary; biro.krisztina@med.unideb.hu; 3Institute of Healthcare Industry, University of Debrecen, Nagyerdei Sqr 98, 4032 Debrecen, Hungary; 4Hospital Pharmacy at the University of Debrecen, University of Debrecen, Nagyerdei Sqr 98, 4032 Debrecen, Hungary

**Keywords:** nasal filters, individual protective devices, infective respiratory diseases

## Abstract

Nowadays, in addition to diseases caused by environmental pollution, the importance of personalized protection against various infectious agents has become of paramount importance. Besides medicine, several technical and technological studies have been carried out to develop suitable devices. One such revolutionary solution is the use of personalized nasal filters, which allow our body to defend itself more effectively against external environmental damage and pathogens. These filters are small devices that are placed in the nose and specifically filter the inhaled environmental contaminants, allergens, and microorganisms according to individual needs. These devices not only play a key role in maintaining our health but also contribute to environmental protection, reducing the inhalation of pollutants and their harmful impact on the natural environment. Another advantage of personalized filters is that they also provide an opportunity to strengthen our individual immune systems. The use of personalized filters allows medicine to provide optimized protection for everyone, focusing on individual genetic and immunological conditions. The momentum behind the development and research of personalized nasal filters has reached astonishing proportions today. Nowadays, many research groups and medical institutions are working to create new materials, nanotechnologies, and bioinformatics solutions in order to create even more effective personalized nasal filters that can also be shaped easily and safely. Considering the needs of the users is at least as important during development as the efficiency of the device. These two properties together determine the success of the product. Industry research focuses not only on improving the efficiency of devices, but also on making them more responsive to user needs, comfort, and portability. Based on all this, it can be concluded that personalized nasal filters can be a promising and innovative solution for protection against environmental pollutants and pathogens. Through a commitment to the research and development of technology, the long-term impact of such devices on our health and the environment can be significant, contributing to improving people’s quality of life and creating a sustainable future. With unique solutions and continuous research, we give hope that in the future, despite the environmental challenges, we can enjoy the protection of our health with even more efficient and sophisticated devices.

## 1. Introduction

The last infectious respiratory disease pandemic resulted in profound changes in human protective behavior worldwide [1]. Social distancing, mask-wearing, hand hygiene, remote work, and vaccination have become integral parts of daily life in the fight against the virus [2]. These behavioral changes have demonstrated the importance of individual and collective responsibility in controlling infectious diseases [3]. While some protective measures may ease as the pandemic is brought under control, others may leave a lasting impact on societal norms and practices [4]. As we navigate the post-pandemic world, the lessons learned from these behavioral changes will likely inform future preparedness for potential outbreaks and other public health challenges [5]. Continued public health education, accessible healthcare, and effective communication between governments and communities will be essential in sustaining a culture of protective behaviors to safeguard public health in the years to come [6]. Respiratory diseases can be more dangerous than other infections due to several key factors, which include their mode of transmission, the vital function of the respiratory system, and the potential for rapid spread within populations [7]. Respiratory diseases can pose significant health risks due to the mode of transmission, the normal vulnerability of the respiratory system, and several other factors [8]. Respiratory diseases primarily spread through respiratory droplets containing infectious agents, such as viruses or bacteria, which can then be inhaled by nearby individuals [9]. This mode of transmission makes respiratory diseases highly contagious and efficient in spreading from person to person, especially in crowded and enclosed spaces. Symptoms characteristic of respiratory infections vary widely [10]. They can range from very mild breathing discomfort to severe breathing difficulties that greatly impair the quality of life [11]. Certain infections can develop rapidly, leading to acute respiratory distress syndrome (ARDS). With ARDS, the lungs become severely inflamed, compromise oxygen exchange, and require mechanical ventilation to support breathing. Respiratory diseases caused by the coronavirus (SARS-CoV), Middle East respiratory syndrome coronavirus (MERS-CoV), or SARS-CoV-2 (COVID-19) have led to serious respiratory problems [12]. These diseases have an extremely high mortality rate. The elderly, young children, and people with chronic diseases are at greatest risk. Many other respiratory diseases, such as influenza and common cold viruses, exhibit seasonal patterns, leading to regular outbreaks [13]. These outbreaks can put a strain on healthcare systems, especially during peak periods when the number of cases surges. While some respiratory infections can be managed with antiviral medications, there may be limited treatment options for specific viral infections. Vaccines against certain respiratory pathogens are already available, but effective vaccines against seasonal or completely new genetic agents may take time to develop. Respiratory viruses, especially RNA viruses such as influenza and coronaviruses, have the ability to mutate rapidly [14]. This mutation can lead to the emergence of new strains with altered virulence or transmissibility, making these vaccine development efforts very difficult. These mutations can challenge existing immunity, making it challenging to develop effective vaccine treatments and protective devices such as individual nasal filters. Besides the infectious agents, we also have to take into account diseases caused by increasing environmental pollution despite all regulatory efforts. Reducing air pollution has long been the focus of attention [15]. Today, however, it has become a major global health challenge, adversely affecting millions of people around the world. The adverse health effects of air pollution are well known, and numerous biochemical, pathological, and epidemiological studies have highlighted the long-term consequences of these effects [16]. Air pollution is a major risk factor for respiratory diseases, including chronic obstructive pulmonary disease (COPD), asthma, and acute respiratory infections [17]. Particulate matter (PM), nitrogen oxides (NOx), sulfur dioxide (SO_2_), and ozone (O_3_) are key contributors to respiratory ailments. Protection is very difficult due to the different particle sizes of harmful substances (Figure 1). Fine particulate matter (PM2.5) is particularly concerning as it can penetrate deep into the lungs and even enter the bloodstream, causing systemic inflammation [18]. Various research results have demonstrated the connection between air pollution and cardiovascular diseases (CVDs), such as heart attack, stroke, and high blood pressure [19]. Exposure to pollutants triggers inflammation, oxidative stress, and autonomic nervous system dysregulation, promoting atherosclerosis and other cardiovascular complications. Long-term exposure to air pollution has been linked to an increased risk of lung cancer and other cancers, including bladder and breast cancer [20]. The carcinogenic pollutants most abundant in polluted air, such as benzene, formaldehyde, and polycyclic aromatic hydrocarbons (PAHs), contribute to these malignancies. Recent research has also shown a link between air pollution and neurological disorders [21]. Fine particles and toxic pollutants can cross the blood–brain barrier, leading to neuroinflammation and neurodegenerative conditions such as Alzheimer’s and Parkinson’s disease. In addition, exposure to prenatal air pollution is linked to adverse birth outcomes such as low birth weight, premature birth, and developmental problems in children [21]. These outcomes are mainly due to the effects of pollutants on placental function and fetal development. According to the abovementioned health-damaging effects, it can be clearly stated that cleaning the air intended for inhalation is not only a health but also an epidemiological issue [22]. In the process of selecting articles for this review, we employed a systematic approach. We conducted comprehensive searches in multiple reputable academic databases, using relevant keywords related to nasal filters and their applications. Our inclusion criteria focused on articles published in peer-reviewed journals, covering the topic of personalized nasal protective devices. We excluded duplicates, conference abstracts, and studies unrelated to the subject. The selected articles were assessed for their relevance, quality, and contributions to the field. This rigorous selection process ensured that the included studies were representative of the most current and informative research in the area of personalized nasal protection.

## 2. The Importance of Individual Nasal Filters in Modern Air Quality Management

Modern air quality management (AQM) is a complex strategic system that focuses on protection against various environmental influences. Any devices with valuable merits can be integrated into AQM (Figure 2). Individual nasal filters are small devices designed to be placed discreetly within the nostrils. These filters act as a physical barrier, preventing airborne particles and pollutants from entering the respiratory system while allowing clean air to pass through [23]. These filters are typically made from hypoallergenic materials, ensuring they are safe for extended use and suitable for individuals with sensitivities [24]. The most significant benefit of individual nasal filters is their ability to protect our respiratory system from various airborne contaminants [25]. These filters effectively trap and block particulate matter, dust, pollen, allergens, smoke, and even certain microorganisms. As a result, they offer a layer of defense against respiratory irritants, reducing the risk of allergies, sinus infections, and other respiratory ailments [26]. By preventing harmful particles from reaching the lungs, individual nasal filters can significantly contribute to improved respiratory health. For individuals who suffer from chronic respiratory conditions like asthma or chronic obstructive pulmonary disease (COPD), using nasal filters can provide added protection and relief from exacerbations triggered by air pollution [27]. One of the unique advantages of individual nasal filters is their personalized approach to air quality management. Unlike standard air purifiers that clean the surrounding environment, nasal filters create a personal clean air zone, which is particularly beneficial in situations where air quality is challenging to control, such as during outdoor activities or when exposed to indoor allergens like dust mites or pet dander [28]. Individual nasal filters offer a practical and convenient solution to maintaining clean air intake. Their compact size and ease of use make them a portable and discreet option for people of all ages. Unlike masks or respirators, nasal filters do not obstruct the face, making them more comfortable for extended wear [29]. Additionally, they are disposable or easily washable, ensuring regular replacement for optimal performance. In the broader context of air quality management, the widespread adoption of individual nasal filters can lead to positive environmental implications. By offering a personalized approach to clean air intake, these filters may reduce the demand for energy-intensive air purifiers and contribute to lowering our carbon footprint [30].

## 3. Nasal Filter Design and Development: Harnessing Technology for Respiratory Health

The ever-increasing concerns about air pollution and its adverse effects on respiratory health have spurred significant advancements in the design and development of nasal filters. Leveraging cutting-edge technologies, researchers and engineers have focused on creating efficient, comfortable, and personalized nasal filters that effectively protect individuals from infections and airborne contaminants [17]. This essay explores the technological approaches employed in nasal filter design and development, highlighting their importance in mitigating respiratory health risks. Material science plays a crucial role in the development of nasal filters. Researchers are continually exploring new materials that are hypoallergenic, biocompatible, and capable of efficiently filtering a wide range of airborne particles [21]. Advances in nanotechnology have enabled the creation of nano-fiber-based filters, offering enhanced filtration efficiency without compromising breathability. These filters can effectively block fine particulate matter, allergens, and even harmful microorganisms [31]. Computational fluid dynamics (CFD) modeling is an essential technological approach used to simulate and analyze the airflow dynamics within the nasal passages [32]. By studying the nasal anatomy and airflow patterns, engineers can optimize the design of nasal filters to ensure proper fit, minimize air resistance, and maximize filtration efficiency. CFD simulations help refine a filter’s geometry and size to create a snug fit, reducing the risk of air leakage around the filter edges [33]. Three-dimensional printing and rapid prototyping have revolutionized the nasal filter design process [34]. These technologies allow for quick and cost-effective iterations of filter designs, enabling researchers to experiment with various shapes, sizes, and materials [35]. With rapid prototyping, engineers can obtain real-world feedback from users and make necessary adjustments before finalizing the design for mass production [36]. Comfort is a critical aspect of nasal filter adoption. Technological approaches focus on creating ergonomic designs that conform to the nasal anatomy, ensuring a secure and comfortable fit for extended wear. Soft and flexible materials are used to prevent discomfort or irritation, and innovative shapes cater to individual nose sizes and shapes to maximize user satisfaction. Advances in sensor technology have opened new possibilities for smart nasal filters [27]. Sensor-embedded filters can monitor air quality in real time, providing users with information about pollutant levels and potential health risks [28]. These data can be transmitted to smartphones or other devices, empowering individuals to make informed decisions about their activities and exposure to polluted environments. As environmental concerns grow, technological approaches in nasal filter design are increasingly focusing on developing biodegradable and eco-friendly filter materials. This ensures that discarded filters do not contribute to plastic waste and pollution [25]. Biodegradable filters, made from sustainable materials, offer a greener alternative while maintaining the same level of filtration efficiency. Chitosan is a biodegradable and biocompatible polymer derived from the shells of crustaceans [37]. Chitosan-based filters have been studied for drug delivery applications due to their mucoadhesive properties, which enable them to adhere to the nasal mucosa and provide sustained drug release. Biodegradable polymeric nanoparticles, such as those made from polylactic acid (PLA) or poly(lactic-co-glycolic acid) (PLGA), have shown promise as carriers for nasal drug delivery [38]. These nanoparticles can encapsulate drug molecules, protecting them from degradation and enabling controlled release upon contact with the nasal mucosa. Biodegradable hydrogels have been investigated as drug carriers for nasal delivery. Hydrogels can provide a sustained release of drugs and offer a biocompatible matrix for drug dispersion and absorption. Silk proteins have been explored as a biodegradable material for drug delivery. Silk-based filters can be engineered to release drugs gradually, making them suitable for sustained drug release applications. Gelatin, derived from collagen, is another biodegradable material that has been studied for nasal drug delivery. Gelatin-based filters can be formulated to control drug release rates and enhance drug absorption [39]. Alginate is a natural polysaccharide derived from seaweed, and it has been investigated for its potential in nasal drug delivery. Alginate-based filters can form gel-like structures, facilitating controlled drug release and improved retention in the nasal cavity [39]. Liposomes are lipid-based vesicles that can encapsulate both hydrophilic and hydrophobic drugs [40]. Biodegradable liposomes have been explored for their potential in nasal drug delivery, as they can improve drug stability and enhance drug penetration through the nasal mucosa. It is important to note that the successful development and commercialization of biodegradable filters for nasal drug delivery depend on various factors, including biocompatibility, drug release kinetics, drug stability, and regulatory considerations. Researchers and pharmaceutical companies continue to explore and optimize these innovative biodegradable filter materials to improve drug delivery efficiency and patient compliance.

## 4. The Operational Principles of Healthcare Nasal Filters: Design Perspectives

The design of healthcare filters plays a crucial role in their efficiency and comfort. One of the fundamental design considerations for healthcare nasal filters is their shape and fit. A well-designed filter should conform to the anatomical structure of the nasal passages to ensure proper coverage and a secure seal. The shape of the filter should allow it to sit comfortably within the nostrils, preventing any air gaps that could compromise its filtration capabilities. Ergonomic design principles are employed to create filters that suit various nasal shapes and sizes, ensuring a universal fit for users. Effective airflow management is critical for the optimal functioning of nasal filters [34]. Filters must strike a balance between maintaining adequate airflow for natural breathing and offering sufficient resistance to block harmful particles. Designers employ engineering techniques to determine the appropriate porosity and surface area of the filter, allowing for efficient air passage while capturing pollutants and allergens [29]. Healthcare nasal filters use a combination of mechanical and electrostatic filtration mechanisms. As air passes through the filter, larger particles are mechanically trapped by the filter’s fibers or porous structure. Simultaneously, smaller particles are attracted to the filter’s surface through electrostatic charges, effectively capturing them and preventing their entry into the respiratory system. The design of the filter material supports these mechanisms by providing the right balance of surface charge and filtration efficiency (Figure 3). Comfort is a critical factor in the successful adoption of healthcare nasal filters [29]. To ensure user compliance, designers prioritize creating filters that are non-intrusive, lightweight, and soft on the nasal passages. The use of hypoallergenic materials reduces the risk of skin irritation and allergic reactions, making the filters suitable for a broader range of users. Proper retention and stability are essential to prevent the filters from dislodging during regular activities. Designers incorporate features such as integrated adhesive strips or support structures to secure the filter in place without causing discomfort or skin irritation [35]. The design should strike a balance between adequate adhesion and easy removal to avoid potential nasal irritation. The design of healthcare nasal filters also considers durability and reusability. By using robust materials and implementing a modular design approach, manufacturers can create filters that withstand repeated use and cleaning [30]. This not only reduces the environmental impact but also ensures cost-effectiveness for users. The operational principles of healthcare nasal filters encompass a comprehensive design approach aimed at optimizing their performance, comfort, and user experience. Through careful consideration of shape, fit, airflow management, filtration mechanisms, comfort, retention, and durability, designers create filters that effectively protect against airborne pollutants and allergens. These design principles continue to evolve with advancements in materials and technology, making healthcare nasal filters an essential tool for promoting respiratory health in various environmental settings.

## 5. The Sustainability Aspects of Healthcare Nasal Filters: Cost and Reusability

Medical nasal filters have witnessed significant growth in the worldwide market in recent years. The increasing awareness of air pollution and respiratory health concerns has driven the demand for these innovative devices [37]. With a focus on mitigating the effects of airborne pollutants and allergens, medical nasal filters have garnered attention from both consumers and healthcare professionals. Financially, the market for medical nasal filters has experienced substantial expansion due to various factors [41]. The global health crisis, which began in 2019, further accentuated the importance of respiratory health, leading to an upsurge in demand for personal protective equipment (PPE) and air filtration solutions [8]. This surge in demand for nasal filters contributed to the growth of companies operating in this sector. Government initiatives to combat air pollution in major cities and industrial areas have also spurred the adoption of nasal filters. As regulations tighten, individuals seek effective solutions to protect themselves from pollution-related health risks, providing a significant market opportunity for medical nasal filter manufacturers [11]. Furthermore, the global healthcare expenditure trend has played a pivotal role in the market’s expansion. Consumers are becoming increasingly health-conscious, which has translated into higher spending on products that promote respiratory well-being [42]. This, in turn, has attracted investments from financial institutions and venture capitalists, fostering the growth of the medical nasal filter industry. Despite the positive outlook, the market faces challenges, such as competition from traditional respiratory protective devices like masks, and potential pricing concerns for some consumer segments. However, companies that can strike a balance between product affordability and performance are likely to gain a competitive edge. One of the primary considerations in sustainability is cost-effectiveness [43]. Healthcare nasal filters must strike a balance between affordability and performance to ensure widespread adoption. Low-cost filters can be accessible to a broader range of individuals, especially in regions with limited resources or high pollution levels [39]. To achieve cost-effectiveness, manufacturers can employ innovative production techniques, streamline supply chains, and use sustainable materials that do not compromise the filters’ effectiveness. Leveraging economies of scale through mass production and strategic partnerships with healthcare providers can also help reduce overall costs. Furthermore, investments in research and development can lead to the creation of long-lasting and durable filters that minimize replacement frequency, thereby reducing the financial burden on consumers and healthcare systems [43]. A key aspect of sustainability is the potential for reusability and recyclability of healthcare nasal filters. Traditional disposable filters generate significant waste and contribute to environmental degradation. By developing reusable filters, we can significantly reduce the environmental impact of this essential medical equipment. Reusable filters are designed to be easily cleanable and durable, offering multiple uses before requiring replacement. This not only reduces waste generation but also proves more cost-effective for users in the long run. Manufacturers can provide proper cleaning guidelines to ensure the filters’ efficacy and prolong their lifespan. Furthermore, the use of recyclable materials in filter construction can enhance their eco-friendliness. Recycling programs can be established to collect used filters, which are then processed to extract valuable materials for manufacturing new filters or other products, minimizing resource depletion and waste accumulation. Promoting sustainability in healthcare nasal filters requires raising awareness among consumers, healthcare professionals, and policymakers [30]. Educating users about the benefits of reusable filters and the importance of responsible disposal can encourage eco-friendly practices. Besides the abovementioned aspects, the possible release of microplastics from disposable nasal filters is a novel environmental concern with far-reaching implications [32]. These microplastics, once released into the environment, can persist for extended periods, infiltrating ecosystems and potentially harming wildlife. Furthermore, the long-term health consequences of microplastic exposure for both the environment and human populations remain a subject of growing concern. As such, addressing this issue is vital to mitigate potential ecological and health risks associated with microplastic pollution. Governments and regulatory bodies can play a crucial role in fostering sustainability in the healthcare industry by incentivizing the development and adoption of eco-friendly nasal filters. This can include offering tax breaks or financial support to companies that prioritize sustainability and incorporate recyclable materials into their products [43]. The sustainability aspects of healthcare nasal filters are critical considerations as we seek to protect respiratory health while minimizing our environmental impact. Cost-effectiveness and reusability are pivotal in ensuring widespread accessibility and reducing waste generation. By investing in research, development, and education, we can foster a more sustainable future where healthcare solutions, including nasal filters, align with environmental preservation goals. Embracing sustainable practices in the healthcare industry is not only an ethical imperative but also a pathway to better public health and a cleaner environment. The most important findings and the most important evaluation aspects had been summarized in Table 1.

## 6. Ensuring Safety in Healthcare Nasal Filters: Manufacturers’ Testing Protocols

Manufacturers apply rigorous testing protocols to guarantee the safety and efficacy of medical nasal filters, which include various aspects such as the biocompatibility of materials, filtration efficiency, and user comfort. The materials used in healthcare nasal filters must be biocompatible to avoid adverse reactions and skin irritations in users [44]. Manufacturers conduct biocompatibility tests following international standards, such as ISO 10993, which evaluate the compatibility of filter materials with living tissues. These tests assess the potential cytotoxicity, sensitization, and irritation effects of the materials, ensuring that filters are safe for prolonged contact with the nasal passages [45]. The primary function of healthcare nasal filters is to effectively capture and remove airborne particles. To assess their filtration efficiency, manufacturers conduct performance tests using standardized particles of varying sizes. By measuring a filter’s ability to remove specific particle sizes, manufacturers ensure that the filters meet high standards in protecting users from harmful airborne contaminants. Healthcare nasal filters must strike a balance between filtration efficiency and providing sufficient airflow for comfortable breathing [46]. Manufacturers conduct airflow and pressure drop tests to evaluate the resistance the filter poses to air passage. These tests ensure that the filter design allows for unimpeded airflow while maintaining effective filtration, preventing any discomfort or breathing difficulties for users. The structural integrity and durability of healthcare nasal filters are critical to their long-term effectiveness and safety. Manufacturers subject filters to mechanical tests that simulate real-world usage conditions, including bending, stretching, and impact tests. These evaluations ensure that the filters remain durable and reliable throughout their intended lifespan, offering continuous protection to users. For reusable healthcare nasal filters, manufacturers provide guidelines on proper cleaning and maintenance. Prior to market release, filters undergo cleaning and reusability tests to verify their performance before and after cleaning. These tests ensure that a filter’s efficacy remains intact after multiple cleaning cycles, promoting user safety and cost-effectiveness. In environments where bacterial or fungal contamination may be a concern, manufacturers conduct microbial testing on the filter materials. These tests assess filters’ ability to resist microbial growth and prevent potential infections, ensuring users’ safety in various settings. The safety of healthcare nasal filters is a top priority for manufacturers, and they adhere to stringent testing protocols to ensure their efficacy and user well-being. Through material biocompatibility tests, filtration efficiency assessments, airflow evaluations, and durability testing, manufacturers ensure that these filters effectively protect users from airborne pollutants while maintaining user comfort [47]. Compliance with international standards and continuous research contribute to the ongoing improvement of healthcare nasal filters, making them indispensable tools in promoting respiratory health and overall well-being.

## 7. Authorization Process for Placing Products on the Market

Access to and use of medical devices in the European Union market requires a strict licensing process [48]. This ensures the protection of patients and public health, as well as the highest quality and safety of the products. In the European Union, medical devices are licensed based on harmonized regulations, which ensure consistency and uniform standards between member states [49]. The first step in the authorization process is the classification of medical devices. In the EU, the Medical Devices Regulation (MDR) and the In Vitro Diagnostic Devices Regulation (IVDR) govern the classification of devices and the authorization process. Assets are classified based on the risks they carry. The highest-risk devices, such as implants, surgical devices, and IIb and III devices belonging to the class can obtain the CE marking with the involvement of a third party and the cooperation of the notified body [50]. Notified bodies are the authorities of each member state designated to evaluate and authorize devices. They are also required when licensing medical devices. The purpose of these tests is to prove the product’s safety and effectiveness under real conditions. During clinical trials, data are collected on patient tolerance, product performance, and possible side effects. These data are of crucial importance during the authorization of the device [50]. One of the important steps in the authorization procedure is the compilation of the technical documentation, which contains the design and technical data of the device, the results of clinical tests, as well as quality assurance and production information. Based on this documentation, the notified body evaluates the compliance of the device with the regulations. Before medical devices are placed on the market, manufacturers must prepare product labeling, instructions for use, and information materials for patients. This information must be clear and understandable to users. To enter the single European market, medical devices must meet the identified standards and requirements of the countries of the European Economic Area. The authorization process can be complex, time-consuming, and expensive, but it is essential to guarantee the security and efficiency of devices [51].

## 8. Combination of Nasal Filters with Drug Delivery Devices

The combination of personal nasal filters and pharmaceutical nasal dispensers presents an innovative approach to respiratory health management [52]. By integrating preventive measures with targeted medication delivery, this hybrid solution offers enhanced protection, improved medication efficacy, and extended medication lifetime (Figure 4). As technology continues to advance, further research and development in this area could open new possibilities for respiratory care, ultimately contributing to better health and quality of life for individuals around the world [28]. Combining personal nasal filters with pharmaceutical nasal dispensers can provide dual protection. Nasal filters act as the first line of defense, preventing allergens and pollutants from entering the nasal passages. Simultaneously, pharmaceutical nasal sprays can address existing respiratory issues by delivering medication directly to affected areas, providing prompt relief. Nasal filters can enhance the effectiveness of pharmaceutical nasal sprays by reducing the presence of external irritants and pollutants in the nasal cavity [29]. Cleaner nasal passages may allow medications to better reach their target sites, potentially improving their overall efficacy. Combined devices might be able to extend the medication lifetime. Nasal filters can help minimize contact between nasal tissues and the drug particles in the spray. This reduction in direct contact may lead to decreased drug degradation, potentially extending the shelf life and effectiveness of the medication. Moreover, this possibility might result in enhanced patient compliance [53]: The combination of personal nasal filters and pharmaceutical nasal dispensers provides a comprehensive approach to respiratory care [54]. By addressing both preventive and therapeutic aspects, this approach may foster greater motivation among individuals to adhere to their treatment plans, resulting in improved overall compliance [55].

## 9. Discussion

In our approach to article selection for this review, we followed a systematic procedure. We conducted thorough searches across reputable academic databases, utilizing pertinent keywords pertaining to nasal filters and their applications. Our inclusion criteria centered on articles published in peer-reviewed journals, specifically addressing personalized nasal protective devices. Modern medicine and engineering innovation are more and more frequently setting common goals, thereby creating new opportunities, for example, in the field of patient care and drug administration. One of the exciting developments is the concept of personalized nasal filters, which can revolutionize healthcare in many areas, from the prevention of respiratory diseases to the targeted intake of medicines. Personalized nasal filters are small devices that, when placed in the nasal passage, filter and clean the air of harmful particles and allergens. These filters can not only play a key role in improving air quality, but can be customized according to individual characteristics, so they can provide effective protection for people with allergies, asthma, and respiratory diseases. The personalized approach allows the tools to be adapted to the specific needs and circumstances of the individual. In addition, the design and development of personalized nasal filters has also opened the door to the targeted administration of drugs. The nasal passage is extremely sensitive and quickly absorbs active ingredients, so it can be an ideal channel for the effective and rapid introduction of medicines into the body. Personalized filters allow drugs, such as pain relievers or hormone treatments, to reach precise target areas in the body, minimizing common side effects and dosing issues. Before medical use, however, research and development processes are necessary so that the effectiveness and safety of the devices can be properly verified. The cooperation of engineers and doctors is fundamental in the development and application of personalized nasal filters. During clinical trials, the effectiveness of the filters in protection against air pollution and allergens, as well as in the effective intake of medicines, must be evaluated. In addition, the individual reactions and effects of the filters must also be examined when applying them to different patient groups. The intersection of healthcare and engineering innovation in medical devices has always brought significant advances. Personalized nasal filters are not just a new technology, but a whole new approach to health protection and drug administration. These devices can improve both patient care and patients’ quality of life, as well as open new horizons in the delivery of more effective drugs. However, for personalized nasal filters to become a real breakthrough in healthcare, continuous research, development, and clinical evaluation are essential. In the future, these devices may open a new dimension in healthcare and treatment, and contribute to a better quality of life for patients. The future of nasal filter development holds promise and presents exciting possibilities in the realm of personal respiratory protection. Forward-looking research is geared towards enhancing filter efficiency by designing ultrafine filters capable of more effectively blocking harmful particles and allergens. Active filter technologies are also a focal point for development, enabling filters to actively respond to various environmental challenges, such as air pollution. Sustainability remains a crucial concern, with the use of organic and recycled materials on the rise, along with the design of recyclable filters. The increasing adoption of IoT (Internet of Things) applications and smart devices facilitates the easy monitoring of filter performance and replacement. In the future, there will be growing opportunities in the realm of personalized filters. The continued growth of applications in healthcare and sports provides space for filters that can be customized and optimized to individual needs. However, several challenges must be addressed. Striking a balance between simplicity and effectiveness, filters need to offer enhanced protection against environmental challenges. Harmonizing technological advancements with sustainability principles holds the potential for revolutionary innovations in nasal filters, allowing for broader applications in public health and everyday life in the coming decades.

## Figures and Tables

**Figure 1 life-13-02116-f001:**
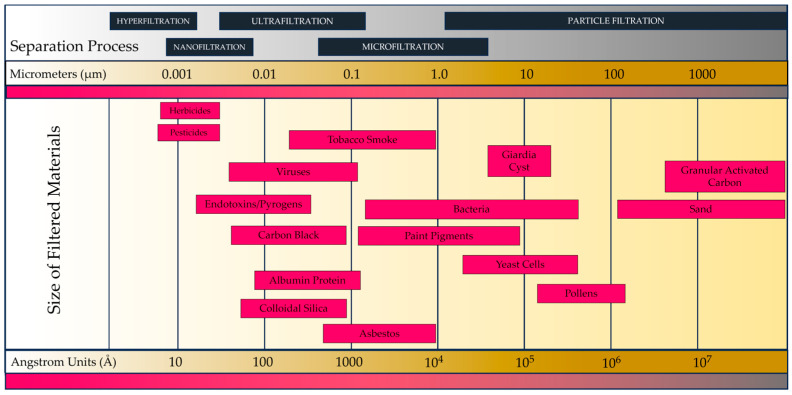
Sizes of filtered materials (µm and Å) and possible separation processes.

**Figure 2 life-13-02116-f002:**
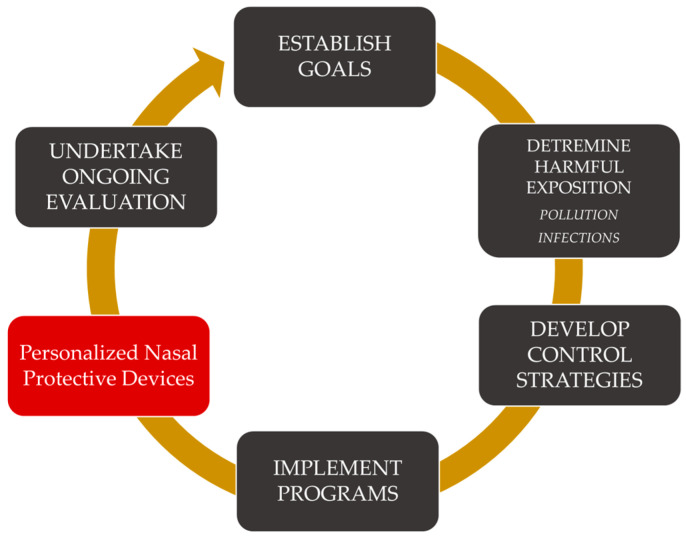
Implementation of personal nasal filters in AQM systems.

**Figure 3 life-13-02116-f003:**
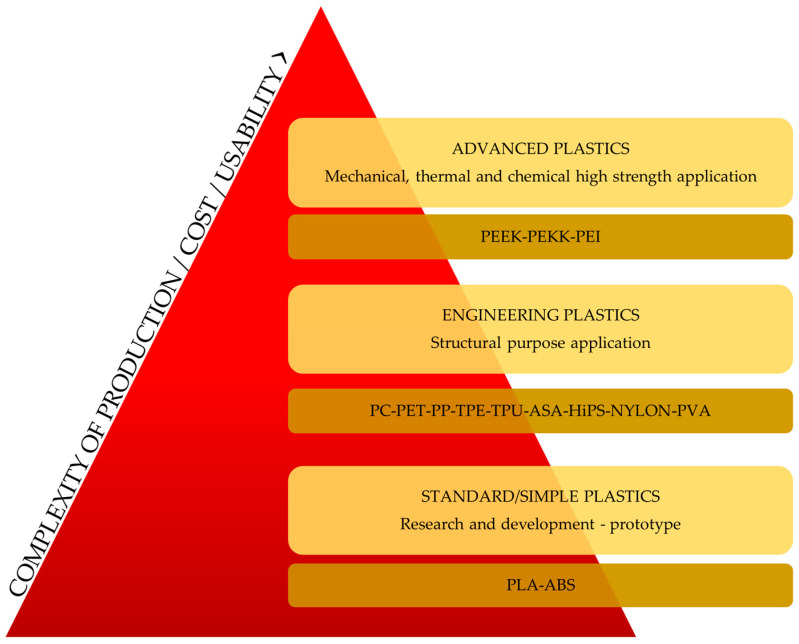
Classification of filaments for 3D technology, based on their applicability and cost.

**Figure 4 life-13-02116-f004:**
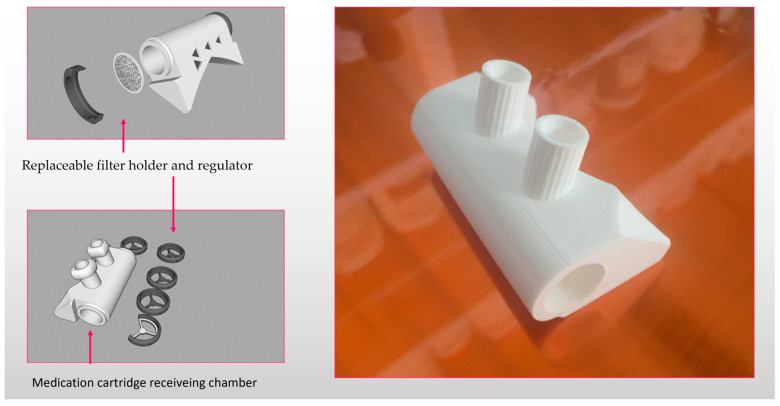
FDM printed prototype of a combined nasal drug dispenser for solid nanoparticles [53].

**Table 1 life-13-02116-t001:** Summary of key findings, materials, filter types, applications, prices, sustainability aspects, and limiting factors.

Key Findings	Materials Used	Types of Filters	Application Areas	Prices	Sustainability Aspects	Limiting Factors
Effective Allergen Filtration	Polypropylene	Mechanical Filters	Allergy Management	Moderate Range	Recyclable materials	Limited efficacy with ultrafine particles
Environmental Protection	Activated Carbon	Electrostatic Filters	Respiratory Health	Low Range	Low environmental impact	Short filter lifespan
Microorganism Blockage	Silicone	Nano-fiber Filters	Healthcare Applications	High Range	Antibacterial properties	Potential skin irritations
Hypoallergenic Materials	Cotton	Carbon Dioxide Filters	Pediatric Respiratory	Variable Pricing	Organic and hypoallergenic materials	Reduced filtration efficiency
Comfortable Fit and Wear	Polyester	Active Filters	Sports and Recreation	Various Price Ranges	Enhanced breathability	Limited protection against gases
Durability and Reusability	Plastic	Shrink-wrapped Filters	Workplace Environment	Affordable Options	Reusable and sustainable	Reduced filtration capacity over time

## Data Availability

No new data were created or analyzed in this study. Data sharing is not applicable to this article.

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
