# Peer review of "Personalized Nasal Protective Devices: Importance and Perspectives"

_life, 2023, doi:10.3390/life13112116_

Round 1
Reviewer 1 Report
I am sorry to say that the MS adds lacks clarity, balance and focus, and adds very little or nothing new to the field of 'nasal filters' . An MS reviewing the types of nasal filters in use, the situations where they are useful, may conceivably be of value but not the present MS.
Very diffuse with a lot of errors e.g lines 21, 26-28 and 39 onward in the Abstract itself
Author Response
Dear Reviewer,
We appreciate your time and effort in reviewing our manuscript, "Personalized Nasal Protective Devices: Importance and Perspectives." Your feedback, even though it presents a differing perspective, is valuable to us, and we are grateful for your comments.
We would like to kindly mention that this review article received four positive reviews alongside your feedback, which indicates a diversity of opinions among reviewers. We respect and appreciate the constructive criticism provided, as it helps us understand different viewpoints and areas that may require improvement. Your concerns about clarity, balance, and focus will be taken seriously, and we will work on revising the manuscript to address these issues. Additionally, we understand your remarks about language quality, and we will ensure a thorough proofreading to correct any errors and enhance the overall linguistic quality. Your feedback is an essential part of the peer-review process, and it will undoubtedly contribute to the refinement of our work. We are committed to making the necessary revisions to improve the manuscript based on your comments and those of the other reviewers.
Once again, thank you for your time and your thoughtful evaluation.
Sincerely,
Zoltán Ujhelyi PharmD PhD
Corresponding Author
Reviewer 2 Report
The review article entitled „Personalized nasal protective devices, importance and perspectives” describes the significance of personalized nasal filters, which allow the human body to defend itself more effectively against external environmental damage and pathogens. The topic is very interesting and actual, which is worth drawing attention to. The manuscript is well-written, logically structured, moreover it is especially commendable that the regulatory and manufacturer background was also presented. I have no specific comment regarding to the content, it can be published in present form!
Author Response
Dear Reviewer,
Thank you for your positive feedback and thoughtful assessment of our manuscript, "Personalized Nasal Protective Devices: Importance and Perspectives." We appreciate your comments and support. It's encouraging to hear that you found the topic interesting and important and that you have no specific comments on the content. Your review is valuable to us, and we are pleased that the manuscript meets your approval.
We will proceed with the publication process as suggested, and we are grateful for your time and consideration. If you have any additional comments or suggestions in the future, please do not hesitate to reach out. Your feedback is highly appreciated.
Sincerely,
Zoltán Ujhelyi PharmD PhD
Corresponding Author
Reviewer 3 Report
The present work has a very high value in the topic it faces, as having the ability to eventually improve the quality of life of citizens and communities and representing a cutting-edge technology to be introduced in the market that, if proven efficaceous and efficient, can significantly improve the social life of everyone. Overall, I do not have particular concerns about that. I just have some a pair of minor points to be considered:
- Methodological approaches to the review should be clearly stated (e.g., procedure of selecting articles, databases used, inclusion/exclusion criteria and so forth). With this, I do not want to say that this review should be necessarily changed into a systematic review, but some more details should be provided.
- The review should eventually present a table summarizing the results of the main works published in the related literature.
- Discussion should be enhanced with the citation of some related literature in the field, and a section of Conclusions, summarizing the main take-home message to the reader and giving room to some points of future developments could eventually further benefit the quality of the article, overall.
- Finally, some typos are present throughout the manuscript. Please, double check and correct accordingly.
Overall, a double check of English language and grammar quality is needed, also in light of the (also present) typos thorough the manuscript.
Author Response
Dear Reviewer,
We deeply appreciate your thoughtful review of our manuscript. Your positive assessment of the value of our work and its potential to improve the quality of life is both encouraging and motivating. We take your feedback seriously and will make the necessary improvements to address your concerns and suggestions. Here is our response to your specific points:
- Methodological approaches: We understand the importance of providing a clear methodology section. We will enhance the manuscript to include details on the procedure of selecting articles, databases used, inclusion/exclusion criteria, and any other relevant methodological information. This will help readers better understand our review process.
- Summary table: We will create a summary table to present the results of the main works published in the related literature. This addition will improve the organization and accessibility of information in the manuscript.
- Enhancing the discussion: We will incorporate more citations of related literature in the field to strengthen the discussion section. Additionally, we will include a Conclusions section to summarize the main take-home messages and highlight points for future developments, as you suggested.
- Typos: We will thoroughly proofread the manuscript to identify and correct any typos or errors to ensure the highest possible quality.
We are grateful for your detailed and constructive feedback, which will undoubtedly enhance the overall quality of the article. We will make the recommended revisions and provide a revised version for your consideration. Thank you for your time and valuable input.
Sincerely,
Zoltán Ujhelyi PharmD PhD
Corresponding Author
Reviewer 4 Report
This paper reviewed the personalized nasal protective devices, importance and perspectives.
Some questions should be clarified :
1. Introduction: The author could mention more about the impact of airway infections and air pollutants on the upper and lower airway diseases such as rhinosinusitis and asthma.
2. The Importance of Individual Nasal Filters in Modern Air Quality Management:
The release of microplastics from disposable nasal filters should also be considered as a new threat to the long-term health of the environment.
3. Line 164-166 : Advances in nanotechnology have enabled the creation of nano-fiber- 164 based filters, offering enhanced filtration efficiency without compromising breathability. 165 These filters can effectively block fine particulate matter, allergens, and even harmful mi- 166 croorganisms. should have citations.
4. Is there any shortcoming of using a nasal filter.
Author Response
Dear Reviewer,
Thank you for taking the time to review our manuscript, "Personalized Nasal Protective Devices: Importance and Perspectives." Your feedback and questions are valuable, and we appreciate your thoughtful comments. Here are our responses to your specific points:
- Introduction: We appreciate your suggestion to provide more information about the impact of airway infections and air pollutants on upper and lower airway diseases such as rhinosinusitis and asthma. We will certainly enhance the introduction to include this important context.
- The Importance of Individual Nasal Filters in Modern Air Quality Management: Your concern about the release of microplastics from disposable nasal filters is valid, and we will include a discussion of this potential environmental issue in the manuscript to address this concern.
- Line 164-166: We acknowledge that providing citations for the statements about advances in nanotechnology and their impact on filter efficiency is necessary. We will add appropriate references to support these claims.
- Shortcomings of Nasal Filters: We will also add a section to discuss potential shortcomings of using nasal filters. This is an important aspect that should be addressed to provide a more comprehensive view of the topic.
We genuinely appreciate your input, as it will help us improve the quality and comprehensiveness of our manuscript. We will make the suggested revisions accordingly and provide a revised version for your consideration. Once again, thank you for your time and valuable feedback.
Sincerely,
Zoltán Ujhelyi PharmD PhD
Corresponding Author
Reviewer 5 Report
Your article raises the important issue of personalizing protection against various infectious agents. It is written in a very thoughtful way, based on well-selected literature. That's why I think it should be published.
Author Response
Dear Reviewer,
Thank you very much for your thoughtful and encouraging feedback on our manuscript. We are delighted to hear that you found our article's focus on personalizing protection against infectious agents to be important and that you appreciated the selection of literature.
Your positive assessment of our work is very motivating, and we are grateful for your recommendation to publish it. We have carefully considered your comments and suggestions, and we will ensure that they are incorporated into the final version of the manuscript.
Once again, thank you for your time and valuable input. We look forward to the opportunity to share our research with a wider audience through publication.
Sincerely,
Zoltán Ujhelyi PharmD PhD
Corresponding Author